# Fabrication of Superhydrophobic/Superoleophilic Bamboo Cellulose Foam for Oil/Water Separation

**DOI:** 10.3390/polym14235162

**Published:** 2022-11-27

**Authors:** Chun-Hua Liu, Jiao-Ping Shang, Xing Su, Shuang Zhao, Yun Peng, Yi-Bao Li

**Affiliations:** Engineering Research Center of Jiangxi Province for Bamboo-Based Advanced Materials and Biomass Conversion, College of Chemistry and Chemical Engineering, Gannan Normal University, Ganzhou 341000, China

**Keywords:** wetting, bamboo cellulose, cellulose foam, superhydrophobic/superoleophilic, oil/water separation

## Abstract

Water is an indispensable strategic resource for biological and social development. The problem of oily wastewater pollution originating from oil spillages, industrial discharge and domestic oil pollution has become an extremely serious international challenge. At present, numerous superwetting materials have been applied to effectively separate oil and water. However, most of these materials are difficult to scale and their large–scale application is limited by cost and environmental protection. Herein, a simple, environmentally friendly strategy including sol–gel, freeze–drying and surface hydrophobic modification is presented to fabricate a bamboo cellulose foam with special wetting characteristics. The bamboo cellulose foam is superhydrophobic, with a water contact angle of 160°, and it has the superoleophilic property of instantaneous oil absorption. Owing to the synergistic effect of the three–dimensional network structure of the superhydrophobic bamboo cellulose foam and its hydrophobic composition, it has an excellent oil–absorption performance of 11.5 g/g~37.5 g/g for various types of oil, as well as good recyclability, with an oil (1,2–dichloroethane) absorption capacity of up to 31.5 g/g after 10 cycles. In addition, the prepared cellulose–based foam exhibits an outstanding performance in terms of acid and alkali corrosion resistance. Importantly, owing to bamboo cellulose being a biodegradable, low–cost, natural polymer material that can be easily modified, superhydrophobic/superoleophilic bamboo cellulose foam has great application potential in the field of oily wastewater treatment.

## 1. Introduction

The ecological environment on which mankind and all living beings on Earth depend for survival has been seriously damaged due to oily wastewater pollution resulting from the discharge of oil spill accidents, and industrial and domestic oily wastewater. Ref. [1] Thus, the effective separation of an oil–water mixture is now regarded as a major challenge by researchers. Refs. [2,3] A comparison of traditional methods, including incineration and biodegradation, Ref. [4] filtration and adsorption are considered as effective technologies to solve the problem of oil–water separation. Refs. [5,6,7,8,9] The filtration method refers to oil or water selectively passing through a filter material, while the other phase is retained, to achieve oil–water separation, Refs. [10,11] such as via the use of a superhydrophobic metal mesh, Refs. [12,13,14,15] superwetting organic polymer, Refs. [16,17,18,19] hierarchical inorganic materials [20,21,22] and more. The adsorption technique is used to selectively adsorb oil or water to the surface and internal voids of the adsorptive material and prevent the infiltration of another phase, Ref. [23] for instance, via the use of superhydrophobic melamine foam, Refs. [24,25,26] superhydrophobic hybrid sponge [27,28,29] and cellulose sponge [30,31,32]. For the filtration method, membrane plugging caused by high–viscosity oil hinders its practical application. Refs. [33,34,35] In contrast, adsorptive material can realize the separation of oily wastewater with a high viscosity. Therefore, the design and fabrication of adsorption materials with a hierarchical structure and superwetting abilities that are environmentally friendly will provide new ideas for solving the problem of oily wastewater pollution.

Cellulose is an abundant natural resource with many characteristics that are different from petroleum–based polymers, such as biocompatibility, biodegradability and a low cost. Refs. [36,37] With the deepening research on the physical and chemical properties of cellulose, numerous cellulose–based materials such as nanocrystals, Ref. [38] films, Ref. [39] gels [40] and sponges have become available and are widely used in the textile industry [41,42] in energy catalysis, Refs. [43,44,45,46] biomedicine, Refs. [47,48,49] food chemicals, Refs. [50,51,52] wastewater treatment [53,54,55,56] and other fields. Refs. [57,58,59,60,61] At present, a series of special wetting cellulose–based materials have been prepared by various simple methods [62,63] and applied in the field of oil and water separation. Refs. [64,65] Bamboo cellulose is extracted from natural bamboo, which is biodegradable, has good air permeability, is strong and wear–resistant, etc. Refs. [66,67] Therefore, it is imperative to develop bamboo cellulose degradable adsorptive materials to solve the problem of oil–bearing wastewater pollution.

Herein, a facile and green technique was proposed to construct superhydrophobic/superoleophilic bamboo cellulose foam via low–temperature dissolution in an alkali/urea system, a freeze–drying process and hydrophobic–modification with hexadecyltrimethoxysilane (HDTMS). The obtained bamboo cellulose foam possessed excellent properties such as being lightweight (0.053 g/cm^3^) and having superhydrophobic properties (static water contact angle of up to ~160°, oil contact angle is approximately 0° and acid and alkali resistance). Moreover, we systematically studied the oil absorption performance of the superhydrophobic bamboo cellulose foam via a series of oil absorption experiments. In addition, the high recyclability of bamboo cellulose foam was proven by an oil adsorption experiment of 10 cycles. Furthermore, it was revealed that the essence of the adsorbing oil is derived from the capillary force of cellulose fibers with a three–dimensional network structure and special wettability. This superhydrophobic bamboo cellulose foam that is sustainable, low–cost and environmentally friendly is presented as a potential candidate in the field of oily wastewater adsorption treatment.

## 2. Materials and Methods

### 2.1. Materials

Bamboo bleached pulp (molecular weight (*Mη*), 1.9 × 10^5^–2.2 × 10^5^, alpha–fiber content: 86–88%) was supplied by Ganzhou Hwagain Co., Ltd. (Ganzhou, China) (Appendix A). Sodium hydroxide (96%, AR), urea (99%, AR) and hexadecyltrimethoxysilane (HDTMS, 99%, AR) were obtained from Shanghai Aladdin Bio–Chem Technology Corporation (Shanghai, China). The 1,2–dichloroethane (99%, AR), t–butanol (TBA, 99%, AR), para–xylene (99%, AR), toluene (99.5%, AR), benzene (99.5%, AR), cyclohexane (99.5%, AR), petroleum ether (AR) and hexane (97%, AR) were purchased from Xilong Chemical Co., Ltd. (Beijing, China). Vacuum pump oil was purchased from the Beijing Sifang Special Oil Factory (API: CS 100, Beijing, China). Peanut oil was obtained from the local supermarket (Unsaturated fatty acid content 80%, Yimengshan Peanut Oil Co., Ltd. Linyi, China). Oil Red O was purchased from Bio–lab Technology Co., Ltd. (Dye content ≥ 75%, Beijing, China). Deionized water was made in the laboratory and used for all experiments and tests. All reagents were used as–received without further purification.

### 2.2. Fabrication of the Superhydrophobic/Superoleophilic Bamboo Cellulose Foam

First, the bleached bamboo pulp was broken using a grinder and dried at 105 °C for 4 h. A 100 mL low–temperature alkaline mixed solution was produced, which consisted of 1.8 g sodium hydroxide, 10 g urea and 88.2 g water, and it was pre–cooled at −35 °C for 30 min in the cold trap. Then, 3 g of broken bamboo pulp was added to the pre–cooling alkaline mixture and stirred for 10 min at high speed, and a 3 wt% homogeneous solution of bamboo cellulose was obtained. Subsequently, the slurry was poured into a beaker and frozen at −20 °C for 2 h in a refrigerator, and a cylindrical bamboo cellulose containing the solvent was constructed. The bamboo cellulose foam was prepared after a series of operations, including a large amount of washing with deionized water (pH = 7), TBA gradient replacement (H_2_O: TBA, 1:1, *v*:*v*, 1 h; 100% TBA, 1 h) and freeze–drying (−40 °C, 48 h). The bamboo cellulose foam was dipped into the ethanol solution consisting of 7 mmol/L HDTMS, and stirred at 100 rpm/min for 8 h at room temperature. Finally, the superhydrophobic/superoleophilic bamboo cellulose foam was successfully prepared and dried at 70 °C for 12 h. The preparation process for the superhydrophobic/superoleophilic bamboo cellulose foam is presented in Figure 1.

### 2.3. Characterization

The hierarchical structures and surface elements of the bamboo cellulose foam were characterized by using a Quanta−450 scanning electron microscopy (SEM, FEI, Gravenhage, Netherlands) configured with energy dispersive X–ray spectroscopy (EDS) operated at 5.0 kV with sputtered Au. The surface chemical composition was performed by an Avatar Fourier Transform Infrared Spectrometer (FT–IR, Nicolet Company, Washington, DC, USA). Thermal gravimetric analyses (TGA) and derivative thermogravimetry (DTG) were carried out on a Q50 thermal analyzer (NETZSCH Corp, Selb, Germany) with heating in air from 25 to 600 °C at 10 °C/min. The X–ray diffraction patterns were recorded using a D8 ADVANCE diffractometer (Bruker, Karlsruhe, Germany) operated at 40 kV and 40 mA with a scanning speed of 5° min^−1^ in the 2*θ* range of 5–40°. The high temperature resistance tests of prepared materials were realized through a CT–946 temperature control heating plate. Contact angles of the resultant samples were measured via a contact angle analyzer (DSA100, KRÜSS, Hamburg, Germany) at ambient temperature using 2 μL of water, and the obtained contact angle was an average value of the five random positions of the tested specimen surface. The optical images and the detailed process of oil adsorption were recorded using a digital camera (D7100, Nikon, Tokyo, Japan).

### 2.4. Setup of Oil Adsorption

The oil adsorption performance of superhydrophobic bamboo cellulose foam was established. Superhydrophobic/superoleophilic bamboo cellulose foam was dipped into a 100 mL mixture solution composed of oil and water with a ratio of 1:1 (*v*:*v*). The weight of the bamboo cellulose foam was recorded before being immersed in oil; then, after complete immersion, the foam was removed and allowed to drain for a few minutes. The saturated bamboo cellulose foam was then transferred to a pre–weighed container and weighed immediately. Then, the above steps were repeated 10 times for the same sponge. The adsorption capacity (*Q*) was calculated using the weight (g/g) of bamboo cellulose foam before and after adsorption as follows:(1)Q=(WAfter−WBefore)WBefore
where *Q* (g/g) is the adsorption capacity, *W*_After_ (g) is the weight after adsorption, and *W*_Before_ (g) is the weight before adsorption.

## 3. Results and Discussion

### 3.1. Surface Micromorphologies and Chemical Compositions

The density of the prepared bamboo cellulose foam was approximately 0.053 g/cm^3^, allowing for it to stand on micro–cilia array structures of the plant (Figure 2a). The obtained cellulose foam had as lightweight character, resulting from the large amount of air trapped in the three–dimensional network structure of the superhydrophobic bamboo foam. The surface morphology of the superhydrophobic bamboo cellulose foam before and after modification was compared by SEM images, as shown in Figure 2b,c. The fiber surface of the bamboo cellulose foam was smooth and irregularly arranged (Figure 2b) without modified HDTMS. In contrast, after hydrophobic modification, a large number of small spherical particles composed of a siloxane reagent were distributed on the fiber surface of the superhydrophobic bamboo cellulose foam (Figure 2c and Appendix A). The experimental results showed that a chemical reaction occurred between the siloxane reagent (HDTMS) and cellulose. Moreover, the FTIR data of the superhydrophobic bamboo cellulose foam further confirmed that the surface composition had changed, as shown in Figure 2d and Appendix A. The characteristic peaks in bamboo cellulose foam O–H stretching vibration occurred at approximately 3340 cm^−1^; the peak at 2900 cm^−1^ corresponded to C–H stretching, the peak at 1610 cm^−1^ was attributed to C=O vibrations and the peak at 103 cm^−1^ denoted C–O telescopic vibrations, appearing in the aerogel before and after HDTMS modification. Furthermore, a new peak at 800 cm^−1^ evidently represented the bamboo cellulose foam modified with HDTMS, and represented the characteristic absorption peak in Si–O–Si (Appendix A). In addition, the EDS result revealed an obvious characteristic peak in a silicon element, further proving that the HDTMS had successfully reacted on the cellulose surface seeing in Figure 2e, and the quality of HDTMS under load also increased with the increase in HDTMS concentration (Appendix A and Appendix A). The thermal behavior and thermal degradation of superhydrophobic bamboo cellulose foam was studied by means of thermogravimetry analyses (TGA) and derivative thermogravimetry (DTG). This clearly showed that the residual amount of after modification slightly increased, as in Appendix A, and the decomposition temperature was also increased compared to before modification. As shown in Figure 2f, the decomposition temperature was improved from 324 °C to 331 °C, as the thermal stability of the Si–O bond formed after modification is higher than that of the C–O bond before modification. For the XRD spectrum data, as shown in Figure 2g, there are similar characteristic peaks before and after modification and without new crystal planes. The result demonstrated that the hydrophobic modification reacting on the cellulose surface, and cellulose kept its original crystalline form. Moreover, the XRD spectrum exhibited the coexistence of cellulose I and II, mainly because of the bamboo cellulose foam infiltrated in alkaline conditions for a long time.

### 3.2. Wettability and Corrosion Resistance

As shown in Figure 3a, a 10 μL water droplet remained spherical on the super hydrophobic bamboo cellulose foam, while a small oil droplet infiltrated into the porous structure of the foam. This occurred due to the three–dimensional network structure of bamboo cellulose foam, and the hydrophobic chemical composition was determined by the surface wettability. This conforms to the bionic micro–nano binary cooperation strategy. Moreover, the contact angle of oil in air is approximately 0° (Figure 2b) and the contact angle of water is 160° (Figure 2c), and the resultant foam possessed the special wetting properties of superhydrophobicity and superoleophilicity. Significantly, as shown in Figure 3d, three small 10 μL water droplets with different pH values, including pH = 1, (HCl), pH = 7 (H_2_O) and pH = 13 (NaOH), remained spherical on the surface of the superhydrophobic bamboo cellulose foam and did not penetrate it. This result confirmed that bamboo cellulose foam with superwettability has excellent acid and alkali resistance, which is conducive to its application under extreme environmental conditions. Stable superhydrophobic properties are the precondition of cellulose–based material application. As shown in Figure 3e, the prepared superhydrophobic bamboo cellulose foam maintained a contact angle of 158° after 180 days in air, showing long–term hydrophobic stability. Moreover, the high–temperature resistance of superhydrophobic foam is shown in Figure 3f. The contact angle slightly decreased with the increase in temperature, but remained superhydrophobic, with a contact angle of 150°, after heating at 280 °C for 30 min. However, most of the fibers on the superhydrophobic surface were carbonized, and the mechanical strength decreased.

### 3.3. Performance of Oil Adsorption

The hydrophobic–modified bamboo cellulose foam achieved special wettability, which was used to selectively remove oil (heavy oil and light oil) from water. Thus, the mixture separation of oil on water and underwater oil were easily achieved. The resultant bamboo cellulose foam demonstrated a good absorption capacity, as shown in Figure 4a,b. In order to further investigate the absorption performance of the prepared bamboo cellulose foam, various types of oil (e.g., toluene, benzene, paraxylene, hexane, cyclohexane, petroleum ether, pump oil, peanut oil and 1,2–dichloroethane) were used as adsorbates, as they are known freshwater pollutants. The absorption ratio (g/g) is defined as the weight ratio of the adsorbate to the dried bamboo cellulose foam. The bamboo cellulose foam showed good absorption capacity for all tested oils. The absorption ratios were 24 g/g, 17 g/g, 28 g/g, 18 g/g, 25.5 g/g,19.4 g/g,11.5 g/g and 14.8 g/g for toluene, benzene, paraxylene, n–hexane, cyclohexane, petroleum ether, pump oil and peanut oil, respectively, as shown in Figure 4c. Significantly, the adsorption rate for 1,2–dichloroethane was up to 37.5 g/g and was slightly higher than that previously reported for similar cellulose–based foams. Refs. [68,69] The adsorption capacities of the different oily substances were discrepant, resulting from the different density and viscosity of the oily substances. Cellulose–based superhydrophobic/superoleophobic materials with excellent selective adsorption demonstrate good reusability. The oil absorption capacity of bamboo cellulose foam after 10 cycles was tested using 1,2–dichloroethane as an example. As shown in Figure 4d, the superhydrophobic bamboo foam retained a high oil absorption capacity (31.5 g/g) after 10 cycles. Therefore, the results showed that the bamboo cellulose foam with superhydrophobic/superoleophilic characteristics retained a high adsorption efficiency and demonstrated an excellent recycling performance.

### 3.4. Mechanism of Oil Adsorption

As shown in Figure 5, taking toluene, with a lower density than water, as the oil phase, the mechanism of oil–water separation using superhydrophobic bamboo cellulose foam is revealed. During the experiment, the toluene solution constituted the upper layer of the aqueous solution as the density of toluene is lower than water, preventing it from diffusing with water (Figure 5a). As the superhydrophobic/superoleophilic cellulose foam was immersed in the mixed solution, the foam first contacted the oil phase and adsorbed the oil (Figure 5b). The driving force for the adsorption of oil originated from the capillary force of the cellulose fibers with hierarchical structures of superhydrophobic/superoleophilic bamboo cellulose foam. Particles consisting of a siloxane reagent on the surface of bamboo cellulose fibers particualrly increased the surface roughness, magnifying the capillarity and improving the oil–absorption performance.

## 4. Conclusions

In conclusion, based on natural biodegradable bamboo cellulose, a cellulose foam possessing superhydrophobicity and superoleophilicity was proposed for oil–water separation through a versatile method of surface hydrophobic modification. The surface of the prepared cellulose foam was instantaneously penetrated by the oil, while the water was blocked, thereby realizing oil absorption, including underwater and overwater oil. The driving force of the superhydrophobic/superoleophilic cellulose foam in absorbing oil is derived from the capillary force of the superhydrophobic hierarchical structures. Moreover, the results indicate that the resultant bamboo cellulose foam has an excellent oil absorption performance and recyclability. In addition, the superhydrophobic cellulose foam exhibited good acid–alkali resistance, and a high–temperature (280 °C) resistance and stability. This work expands upon the avenues in which the superwetting interface material separates oily wastewater, stimulates new thinking in terms of oil–water separation in harsh conditions, and promotes the development and application of natural bamboo cellulose.

## Figures and Tables

**Figure 1 polymers-14-05162-f001:**
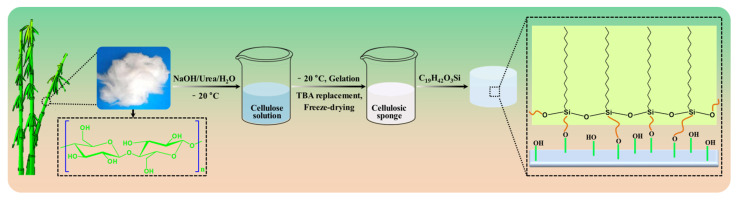
Schematic illustration for the superhydrophobic/superoleophilic bamboo cellulose foam.

**Figure 2 polymers-14-05162-f002:**
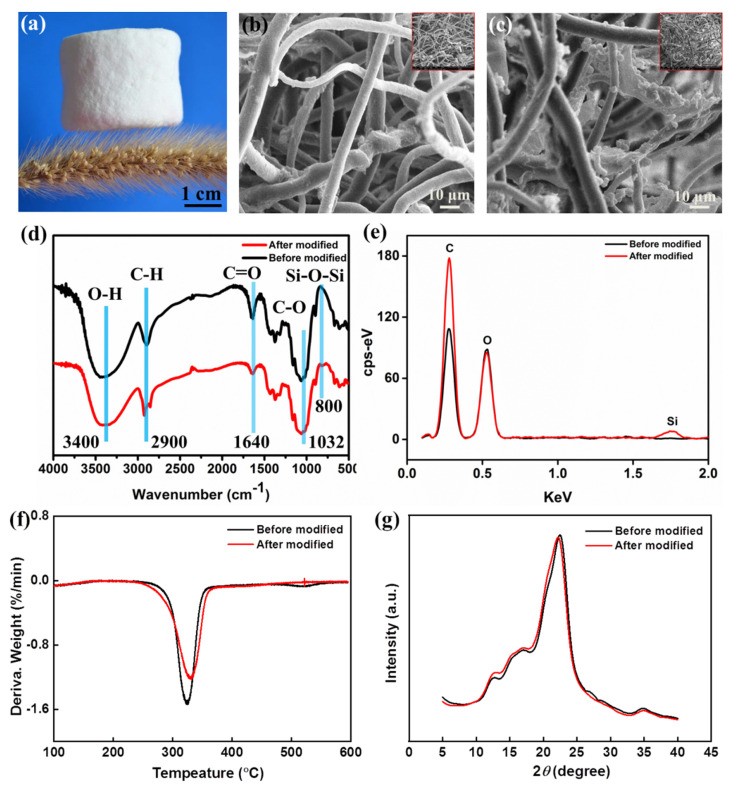
(**a**) Optical image of the superhydrophobic bamboo cellulose foam. SEM images of bamboo cellulose foams before (**b**) and after (**c**) modification. (**d**) FTIR spectra of original foam and superhydrophobic bamboo cellulose foam. (**e**) EDS data of original foam and hydrophobic–modified foam. (**f**) DTG of bamboo cellulose foams before (black line) and after (red line) modification. (**g**) XRD spectrum of bamboo cellulose foams before (black line) and after (red line) modification.

**Figure 3 polymers-14-05162-f003:**
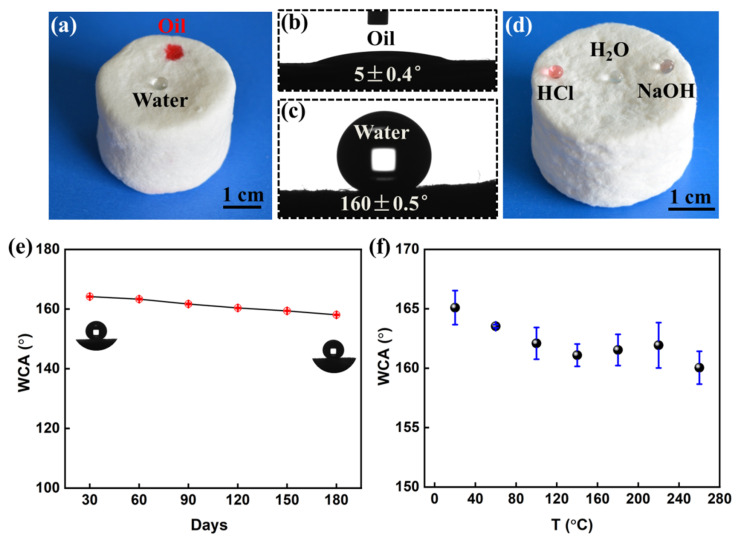
Surface wettability and corrosion resistance. (**a**) Optical photograph of the superhydrophobic bamboo cellulose foam. Contact angle image of oil (**b**) and water (**c**) of the super–wetting bamboo cellulose foam. (**d**) Optical photograph of the acid and alkali corrosion resistance test of superhydrophobic bamboo cellulose foam. (**e**) Influence of time on the stability of superhydrophobic cellulose foam in air. (**f**) High–temperature resistance of superhydrophobic cellulose foam.

**Figure 4 polymers-14-05162-f004:**
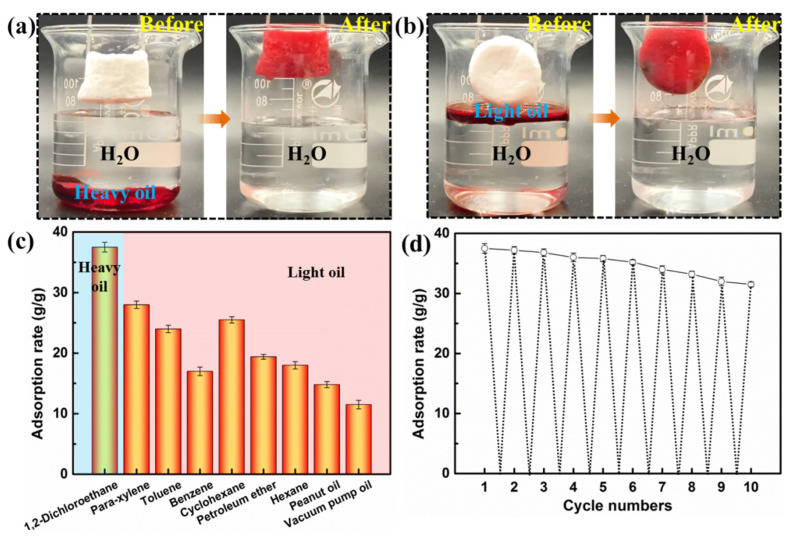
Oil absorption performances. (**a**) Heavy oil (1,2–dichloroethane) adsorption test. (**b**) Light oil (Hexane) adsorption test. (**c**) Diagram of adsorption rate of different oils. (**d**) Number of cycles demonstrating oil (1,2–dichloroethane) absorption performances.

**Figure 5 polymers-14-05162-f005:**
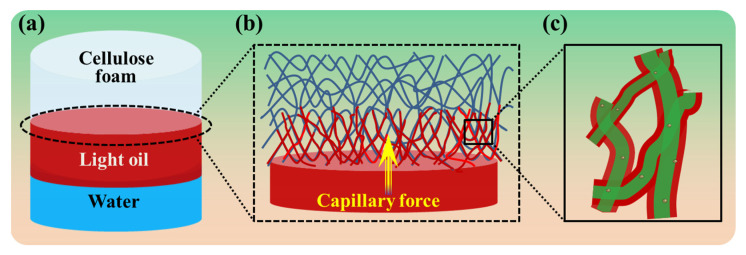
Schematic diagram of oil adsorption of the superhydrophobic/superoleophilic bamboo cellulose foam. (**a**) Absorption process of prepared cellulose foam for light oil. Light oil transport along cellulose fibers under capillary force (**b**) and partial enlarged cartoon (**c**).

## Data Availability

The data presented in this study are available on request from the corresponding author.

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
