# Peer review of "Fabrication of Superhydrophobic/Superoleophilic Bamboo Cellulose Foam for Oil/Water Separation"

_polymers, 2022, doi:10.3390/polym14235162_

Round 1

Reviewer 1 Report

In this manuscript, authors reported the preparation of hexadecyltrimethoxysilane (HDTMS) modified/coated bamboo cellulose foam via sol-gel, freeze-drying and surface hydrophobic modification. The produced foam may be a good potential candidate for oil/water separation. The manuscript presents some relatively good data. I reviewed the manuscript in a critical manner and some of the comments are given below:

General comments

The manuscript might be a contribution of interest for “Polymers” and in principle within its specific scope but it is not suitable for publication in this form. The manuscript is not sufficiently organized with no clear novelty. The quality of writing is average with some grammar and spelling errors and lack of coherence between paragraphs especially those of the introduction section. The English language usage should be checked by a fluent English speaker and/or a professional language editing service.

Moreover, result interpretation is sometimes questionable (e.g. FTIR, mechanism). Some of key characterizations are missing (XRD, TGA, etc)

I recommend major revision.

Specific comments

1.      Many studies in the state of the art used hydrophobization of cellulose with a large range of chemical modifier including siloxane-based modifiers for oil/water separation. Therefore, what is the novelty and the benefit of this work compared with those already reported? I suggest that authors highlight more the novelty in this study.

2.      It seems that the manuscript was written in a very rush manner and wasn’t corrected, check a lot of spelling errors and grammar mistakes throughout the manuscript. For example correct line 15-16 in the abstract ‘’bamboo cellulose foam with superhydrophobicity/superoleophilicity is successfully fabricated’’

3.      The introduction is very poor in terms of the latest advances in cellulose hydrophobization (take a look on Cellulose 28, 7717–7734 (2021). https://doi.org/10.1007/s10570-021-04044-w). More relevant information and data should be given of the already reported works.

4.      What is the weight ratio (wt%) of HDMS coating/modification on the cellulose foam? Was the wt% optimized? For better understanding, authors needs to provide a series of HDTMS coatings with increased wt% to evaluate the influence of the coating content/amount/loading on the contact angle and the oil/water separation performance.

5.      XRD diffractograms and TGA curves need to be provided for all the samples (before and after modification).

6.      In the FTIR/SEM section the author stated ‘’The experimental results showed that the chemical reaction between siloxane reagent (HDTMS) and cellulose occurred’’ what kind of chemical reaction? i.e what is the type of chemical modification? Is it covalent or non covalent? As can be seen from Fig. 1 the authors schematized that HDTMS is covalently bonded with cellulose AGU unit in C6 via Si-O bond. Is possible to establish such covalent modification with such operatory conditions (immersion in ethanol solution containing HDTMS)? More explication needs to be provided to understand the type of modification.

7.      For better understanding of the durability of the highly hydrophobic character, authors needs to evaluate the evolution of water contact angle with time.

8.      Washing tests needs to be provided for the produced samples. Generally, washing test consists of the immersion of the sample in a solution of the washing medium under vigorous stirring for 20 cycles (40 min stirring/cycle). After each cycle, sample is washed, dried and contact the angle value is determined. Different mediums can be used including solvents, hot water (80°C), non ionic detergent, etc.

Author Response

Responses to Reviewer #1

Comments: In this manuscript, authors reported the preparation of hexadecyltrimethoxysilane (HDTMS) modified/coated bamboo cellulose foam via sol-gel, freeze-drying and surface hydrophobic modification. The produced foam may be a good potential candidate for oil/water separation. The manuscript presents some relatively good data. I reviewed the manuscript in a critical manner and some of the comments are given below:

The manuscript might be a contribution of interest for “Polymers” and in principle within its specific scope but it is not suitable for publication in this form. The manuscript is not sufficiently organized with no clear novelty. The quality of writing is average with some grammar and spelling errors and lack of coherence between paragraphs especially those of the introduction section. The English language usage should be checked by a fluent English speaker and/or a professional language editing service. Moreover, result interpretation is sometimes questionable (e.g. FTIR, mechanism). Some of key characterizations are missing (XRD, TGA, etc)

I recommend major revision.

Reply: Thanks for the reviewer’s helpful comment. We have asked the MPDI to polish the language in the whole article, which’s a professional English polishing organization, to improve our paper on the English and grammar usage. Please see if the revised manuscript met the English presentation standard. Moreover, key characterizations are missing (XRD, TGA, etc) have added in the revised manuscript.

Specific comments

  1. Many studies in the state of the art used hydrophobization of cellulose with a large range of chemical modifier including siloxane-based modifiers for oil/water separation. Therefore, what is the novelty and the benefit of this work compared with those already reported? I suggest that authors highlight more the novelty in this study.

Reply: Thanks for the reviewer’s helpful comment. In the previous study, the cellulose-based superhydrophobic materials were fabricated by hydrophobic-modifying with silica based reagents, the fabrication process is complicated. Compared to that work, we have successfully fabricated the superhydrophobic/superoleophilic cellulose using an easy fabrication method (dipping coating). Furthermore, the superhydrophobic cellulose foam exhibited good acid–alkali resistance, and a high temperature (280 °C) resistance and stability, the advantages to have a wide application prospect.

  1. It seems that the manuscript was written in a very rush manner and wasn’t corrected, check a lot of spelling errors and grammar mistakes throughout the manuscript. For example correct line 15-16 in the abstract ‘’bamboo cellulose foam with superhydrophobicity/superoleophilicity is successfully fabricated’’

Reply: Thanks for the reviewer’s comment. We have re-written the abstract section according to the reviewer’s suggestion in the revised manuscript. “Water is an indispensable strategic resource for biological and social development. The problem of oily wastewater pollution originating from oil spillages, industrial discharge and domestic oil pollution has become an extremely serious international challenge. Currently, numerous superwetting materials have been applied to effectively separate oil and water. However, most of these materials are difficult to scale and their large-scale application is limited by cost and environmental protection. Herein, a simple, environmentally friendly strategy including sol–gel, freeze-drying and surface hydrophobic modification is presented to fabricate a bamboo cellulose foam with special wetting characteristics. The bamboo cellulose foam is superhydrophobic with a water contact angle of 160°, and it has the superoleophilic property of instantaneous oil absorption. Owning to the synergistic effect of the three-dimensional network structure of the superhydrophobic bamboo cellulose foam and its hydrophobic composition, it has an excellent oil absorption performance of 11.5 g/g ~ 37.5 g/g for various types of oil, as well as good recyclability, with an oil (1, 2-dichloroethane) absorption capacity of up to 31.5 g/g after 10 cycles. In addition, the prepared cellulose-based foam exhibits an outstanding performance in terms of acid and alkali corrosion resistance. Importantly, owning to bamboo cellulose being a biodegradable, low-cost, natural polymer material that can be easily modified, superhydrophobic/superoleophilic bamboo cellulose foam has great application potential in the field of oily wastewater treatment.”

  1. The introduction is very poor in terms of the latest advances in cellulose hydrophobization (take a look on Cellulose 28, 7717–7734 (2021). https://doi.org/10.1007/s10570-021-04044-w). More relevant information and data should be given of the already reported works.

Reply: Thanks for the reviewer’s comment. According to the reviewer’s helpful suggestion, we have added the relevant contents of the preparation of cellulose-based superhydrophobic materials and the application in the field of oil-water separation.in the revised manuscript. “Currently, a series of special wetting cellulose based materials have been prepared by a variety of simple methods [62-63] and applied in the field of oil and water separation. [64-65]”

  1. What is the weight ratio (wt%) of HDMS coating/modification on the cellulose foam? Was the wt% optimized? For better understanding, authors needs to provide a series of HDTMS coatings with increased wt% to evaluate the influence of the coating content/amount/loading on the contact angle and the oil/water separation performance.

Reply: Thanks for the reviewer’s comment. According to the reviewer’s helpful suggestion, we have performed the additional experiment, and the oil adsorption capacity, coating quality and silicon element content the superhydrophobic cellulose foam are provided in the supporting information. In process of hydrdophobic-modifying bamboo cellulose in various concentrations HDTMS solution including 1 mmol/L, 4 mmol/L, 7 mmol/L, 10 mmol/L and 13 mmol/L, the weight of HDTMS coating increased with the increase of HDTMS solution concentration in following table (Table 1):

Table 1 The changes of after modification foams with different CHDTMS in weight, contact angle and oil absorption

C(HDTMS)

mmol/L

before modification

(g)

after modification

(g)

Δm(After - before)

(g)

adsorption rate

(g/g)

Contact

Angle

(º)

1

0.2135

0.22367

0.01017

11

160.72

4

0.3008

0.31228

0.01148

12.98

163.44

7

0.2407

0.25401

0.01331

11.38

164.39

10

0.2479

0.26452

0.01662

10.75

164.71

13

0.2121

0.23604

0.02394

11.31

165.02

Furthermore, both the silicon element content (Figure S3, Table S2) and contact angle were increased with the increase of HDTMS concentration (Figure 1). Moreover, with the HDTMS concentration increase, the oil absorption amount varied irregularly. The cellulose foam modified in the 4 mmol/L HDTMS solution, the highest adsorption amount is approximatively 13 g/g. This irregular change of oil absorption amount mainly derived from the reaction sites are limited between a single fiber surface (-OH) and HDTMS (-Si-OCH3). Therefore, the results demonstrated that the contact angle and coating quality and silicon element content are increased with the increase of HDTMS concentration, but there is no linear rule for oil adsorption.

Figure 1. The adsorption rate and contact angle of different CHDTMS modification foam

  1. XRD diffractograms and TGA curves need to be provided for all the samples (before and after modification).

Reply: Thanks for the reviewer’s comment. According to the reviewer’s helpful suggestion, we have performed the additional experiments including XRD diffractograms in the revised manuscript Figure 2g, and TGA curves in the revised manuscript Figure 2f and Figure S4.

Figure 2. TGA curves (a), DTG curves (b) and XRD diffractograms (c) of bamboo cellulose foams before (black line) and after (red line) modification

  1. In the FTIR/SEM section the author stated ‘’The experimental results showed that the chemical reaction between siloxane reagent (HDTMS) and cellulose occurred’’ what kind of chemical reaction? i.e what is the type of chemical modification? Is it covalent or non covalent? As can be seen from Fig. 1 the authors schematized that HDTMS is covalently bonded with cellulose AGU unit in C6 via Si-O bond. Is possible to establish such covalent modification with such operatory conditions (immersion in ethanol solution containing HDTMS)? More explication needs to be provided to understand the type of modification.

Reply: Thanks for the reviewer’s helpful comment. We have changed the Figure 1 according to the reviewer’s suggestion in the revised manuscript. According to the literature (Superhydrophobic modification of cellulose sponge fabricated from discarded jute bags for oil water separation, J. Environ. Chem. Eng., 2021, 9(2): 105063), numerous hydroxyl groups (-OH) distributed on C2, C3 and C6 on the surface of natural cellulose. The reaction mechanism of is that the -SiH of HDTMS and -OH of bamboo cellulose form covalent bonds. Firstly, the organosiloxane (HDTMS) is hydrolyzed to produce silicone alcohol, and then the formed silicone alcohol reacted with the -OH (C2, C3 and C6) on the surface of cellulose foam to generate a stable Si-O-Si bond structure. In addition, FTIR spectra of different CHDTMS modification bamboo cellulose foam further confirmed that occur an absorption peak (Si-O-Si) near 800 cm-1 (Figure S2) in the supporting information. It is possible to establish such covalent modification with such operatory conditions (immersion in ethanol solution containing HDTMS).

Figure S2. FTIR spectra of original foam and different CHDTMS modification bamboo cellulose foam

  1. For better understanding of the durability of the highly hydrophobic character, authors needs to evaluate the evolution of water contact angle with time.

Reply: Thanks for the reviewer’s comment. According to the reviewer’s helpful suggestion, the influence of time on the stability of superhydrophobic cellulose foam is provided as Figure 3e in the revised manuscript. The prepared superhydrophobic bamboo cellulose foam still maintained a contact angle of 158° after 180 days in air, showing long-term hydrophobic stability.

Figure 3e (In revised manuscript). Influence of time on the stability of superhydrophobic cellulose foam in air.

  1. Washing tests needs to be provided for the produced samples. Generally, washing test consists of the immersion of the sample in a solution of the washing medium under vigorous stirring for 20 cycles (40 min stirring/cycle). After each cycle, sample is washed, dried and contact the angle value is determined. Different mediums can be used including solvents, hot water (80°C), non ionic detergent, etc.

Reply: Thanks for the reviewer’s comment. According to the reviewer’s helpful suggestion, we have performed the additional experiment, and the stability of the superhydrophobic cellulose foam in a water (pH =7) of the washing medium under vigorous stirring for 20 cycles (40 min stirring/cycle, 800 rpm) are provided, as shown in the following Figure. Owning to the superhydrophobicity of the prepared foam, it floated on the water surface (pH =7) and rotated rapidly under vigorous stirring. The result shown that the foam still retained good superhydrophobicity of after vigorous stirring showed. Compared with the case of acidic (pH =1) and alkaline (pH =13) solution, the superhydrophobic foam rotate rapidly at the initial stage. After 15 minutes of stirring, the superhydrophobic foam sank and slightly damaged, resulting from the acidic/ alkaline solution penetrated into the superhydrophobic foam. Similarly, the stability of surface superhydrophobicity exhibited instability under vigorous stirring in 80 °C hot water and non-ionic detergent (Span 80). However, a l0 μL drops of different alkaline (pH=1), acidic (pH=13), hot water (80 °C) and non-ionic detergents (Span 80) added to the superhydrophobic surface of cellulose foam has a high static contact angle, which proved that the superhydrophobic foam prepared has excellent static superhydrophobic stability. Thanks to the reviewers for their valuable comments, the next work plan will be to fabricate durable superhydrophobic cellulose-based super-wetting interface materials under vigorous stirring.

Figure 3.  Contact angle of superhydrophobic foam in water (pH =7) after vigorous stirring for 20 cycles

Reviewer 2 Report

The manuscript titled "Bioinspired fabrication of superhydrophobic/superoleophilic bamboo cellulose foam for oil/water separation" discusses in detail the synthesis of superhydrophobic/superoleophilic bamboo cellulose foam with excellent oil absorption performance and improved acid and alkali corrosion resistance. The manuscript collectively represents the synthesis, characterization, and applications of bamboo cellulose foam. The manuscript is well arranged and in line with the membranes journal. Furthermore, based on the thoughtful assessment of the manuscript following are the reviewers’ comments:

1.      Kindly, justify the use of the term “Bioinspired” in the title of the manuscript. Else, revise the title.

2.      The reviewer suggests a revision of the abstract avoiding too general discussions and inclusion of (few) mathematical values for the results and other study parameters, for example, oil absorption performance of the bamboo cellulose foam, etc.

3.      The manuscript well presented the performance of the bamboo cellulose foam under different pH conditions. However, the reviewer suggests the inclusion of a discussion about the effect of temperature on the performance of the bamboo cellulose foam. It will be a great addition to the manuscript.

4.      Page No. 5, Line 186, please provide appropriate citations for “…slightly higher than previously reported similar cellulose-based foams.”.

5.      The reviewer thinks the use of the term “super-wetting bamboo cellulose foam” is confusing as it may confuse some of the readers (due to the presence of both superhydrophobic and superoleophilic terms together) as in general this term is used for the complete spreading of water (superhydrophilic) on substrates. Therefore, it can be clarified at the first use of the statement in the manuscript that this term is used for the superoleophilic property of the foam.

Author Response

Responses to Reviewer #3

Comments: The manuscript titled "Bioinspired fabrication of superhydrophobic/superoleophilic bamboo cellulose foam for oil/water separation" discusses in detail the synthesis of superhydrophobic/superoleophilic bamboo cellulose foam with excellent oil absorption performance and improved acid and alkali corrosion resistance. The manuscript collectively represents the synthesis, characterization, and applications of bamboo cellulose foam. The manuscript is well arranged and in line with the membranes journal. Furthermore, based on the thoughtful assessment of the manuscript following are the reviewers’ comments:

1. Kindly, justify the use of the term “Bioinspired” in the title of the manuscript. Else, revise the title.

Reply: Thanks for the reviewer’s comment. According to the reviewer’s helpful suggestion, we have changed the title to “Fabrication of Superhydrophobic/Superoleophilic Bamboo Cellulose Foam for Oil/Water Separation” in the revised manuscript.

2. The reviewer suggests a revision of the abstract avoiding too general discussions and inclusion of (few) mathematical values for the results and other study parameters, for example, oil absorption performance of the bamboo cellulose foam, etc.

Reply: Thanks very much for your comments. We have re-written the abstract section according to the reviewer’s suggestion. In the abstract section, the details of oil absorption performance and cycling performance have added in the revised manuscript. “Water is an indispensable strategic resource for biological and social development. The problem of oily wastewater pollution originating from oil spillages, industrial discharge and domestic oil pollution has become an extremely serious international challenge. Currently, numerous superwetting materials have been applied to effectively separate oil and water. However, most of these materials are difficult to scale and their large-scale application is limited by cost and environmental protection. Herein, a simple, environmentally friendly strategy including sol–gel, freeze-drying and surface hydrophobic modification is presented to fabricate a bamboo cellulose foam with special wetting characteristics. The bamboo cellulose foam is superhydrophobic with a water contact angle of 160°, and it has the superoleophilic property of instantaneous oil absorption. Owning to the synergistic effect of the three-dimensional network structure of the superhydrophobic bamboo cellulose foam and its hydrophobic composition, it has an excellent oil absorption performance of 11.5 g/g ~ 37.5 g/g for various types of oil, as well as good recyclability, with an oil (1, 2-dichloroethane) absorption capacity of up to 31.5 g/g after 10 cycles. In addition, the prepared cellulose-based foam exhibits an outstanding performance in terms of acid and alkali corrosion resistance. Importantly, owning to bamboo cellulose being a biodegradable, low-cost, natural polymer material that can be easily modified, superhydrophobic/superoleophilic bamboo cellulose foam has great application potential in the field of oily wastewater treatment.”

3. The manuscript well presented the performance of the bamboo cellulose foam under different pH conditions. However, the reviewer suggests the inclusion of a discussion about the effect of temperature on the performance of the bamboo cellulose foam. It will be a great addition to the manuscript.

Reply: Thanks for the reviewer’s comment. According to the reviewer’s helpful suggestion, the experiment of high temperature resistance of superhydrophobic cellulose foam is provided as Figure 3f in the revised manuscript. The contact angle has decreased slightly with the increase of temperature, but still remaining superhydrophobicity with a contact angle of 150° after heating at 280 °C for 30 min. However, most of the fibers on the superhydrophobic surface were carbonized, and the mechanical strength decreased.

Figure3f (In revised manuscript).  High temperature resistance of superhydrophobic cellulose foam

4. Page No. 5, Line 186, please provide appropriate citations for “…slightly higher than previously reported similar cellulose-based foams.”

Reply: Thanks for the reviewer’s comment. We have added the related references in the revised manuscript. “Significantly, the adsorption rate for 1,2-dichloroethane was up to 37.5 g/g and was slightly higher than previously reported for similar cellulose-based foams.[68-69]” In references section, “68. Meng, X.; Dong, Y.; Zhao, Y.; Liang, L. Preparation and modification of cellulose sponge and application of oil/water separation. RSC Adv. 2020, 10, 41713-41719. 69. Zheng, X.; Ji, B.; Jiang, R.; Cui, Y.; Xu, T.; Zhou, M.; Li, Z. Polydimethylsiloxane/carbonized bacterial cellulose sponge for oil/water separation. Process Saf. Environ. 2022, 165, 173-180.”

5. The reviewer thinks the use of the term “super-wetting bamboo cellulose foam” is confusing as it may confuse some of the readers (due to the presence of both superhydrophobic and superoleophilic terms together) as in general this term is used for the complete spreading of water (superhydrophilic) on substrates. Therefore, it can be clarified at the first use of the statement in the manuscript that this term is used for the superoleophilic property of the foam.

Reply: Sorry for unclear expression. According to the reviewer’s helpful suggestion, we have provided a clear expression in the revised manuscript, “super-wetting” is revised to "superhydrophobic".

Reviewer 3 Report

Hello dear authors,

The work presented in this paper is quite interesting. 

Here are some comments to improve the quality of the paper.

·       The last paragraph of the introduction must explain the novelty of the work. It is like an abstract.

·       It would be better to present the main achievement of the work by bullets in the conclusion section.

Author Response

Responses to Reviewer #2

Comments: The work presented in this paper is quite interesting. Here are some comments to improve the quality of the paper.

Reply:Thanks for the referee’s positive comment.

  1. The last paragraph of the introduction must explain the novelty of the work. It is like an abstract. It would be better to present the main achievement of the work by bullets in the conclusion section.

Reply: Thanks for the reviewer’s helpful comment. We have re-written the conclusion section according to the reviewer’s suggestion in the revised manuscript. “In conclusion, based on natural biodegradable bamboo cellulose, a cellulose foam possessing superhydrophobicity and superoleophilicity was proposed for oil–water separation through a versatile method of surface hydrophobic modification. The surface of the prepared cellulose foam was instantaneously penetrated by the oil, while the water was blocked, thereby realizing oil absorption, including underwater and overwater oil. The driving force of the superhydrophobic/superoleophilic cellulose foam in absorbing oil is derived from the capillary force of the superhydrophobic hierarchical structures. Moreover, the results indicate that the resultant bamboo cellulose foam has an excellent oil absorption performance and recyclability. In addition, the superhydrophobic cellulose foam exhibited good acid–alkali resistance, and a high temperature (280 °C) resistance and stability. This work expands upon the avenues in which the superwetting interface material separates oily wastewater, stimulates new thinking in terms of oil–water separation in harsh conditions, and promotes the development and application of natural bamboo cellulose.”

Round 2

Reviewer 1 Report

Authors have succesfully answers my questions and fully added my suggestions.

Acceptance is recommended

No further comments.